# Data sharing practices in collaborative human genomic research in low- and middle-income countries: A systematic review protocol

**Deborah Ekusai-Sebatta**[1]*, **Moses Ocan**[2,3], **Shenuka Singh**[4], **David Kyaddondo**[5], **Dickens Akena**[3,6], **Loyce Nakalembe**[7], **Robert Apunyo**[8], **Alison Annet Kinengyere**[1,3], **Eve Namisango**[3,9], **Ekwaro A. Obuku**[3,8,10], **Erisa Mwaka**[1]

1 Department of Anatomy, School of Biomedical Sciences, Makerere University College of Health Sciences, Kampala, Uganda, 2 Department of Pharmacology, School of Biomedical Sciences, Makerere University College of Health Sciences, Kampala, Uganda, 3 Africa Centre for Systematic Reviews & Knowledge Translation, Kampala, Uganda, 4 Discipline of Dentistry, University of KwaZulu Natal, Johannesburg, South Africa, 5 Child Health and Development Centre, School of Medicine, Makerere University College of Health Sciences, Kampala, Uganda, 6 Department of Psychiatry, School of Medicine, Makerere University College of Health Sciences, Kampala, Uganda, 7 Soroti University, Kampala, Uganda, 8 Clinical Epidemiology Unit, Department of Medicine, School of Medicine, Makerere University College of Health Sciences, Kampala, Uganda, 9 Cicely Saunders Institute of Palliative Care, Policy & Rehabilitation Florence Nightingale Faculty of Nursing, Midwifery & Palliative Care, Kings College London University of London, London, United Kingdom, 10 London School of Hygiene and Tropical Medicine, Faculty of Epidemiology and Population Health, London, United Kingdom

* ekusai@gmail.com

**Data Availability Statement:** No datasets were generated or analyzed during the current study. All

## Abstract

### Introduction

The practice of creating large databases has become increasingly common by combining research participants' data into larger repositories. Funders now require that data sharing be considered in newly funded research project, unless there are justifiable reasons not to do so. Access to genomic data brings along a host of ethical concerns as well as fairness and equity in the conduct of collaborative research between researchers from high- income and low-and middle-income countries.

### Materials and methods

This systematic review protocol will be developed in line with PRISMA -guidelines which refers to Open Science Framework, registered in PROSPERO (https://www.crd.york.ac.uk/prospero/) record CRD42022297984 and published in a peer reviewed journal. Data sources will include PubMed, google scholar, EMBASE, Web of science and MEDLINE. Both published and grey literature will be searched. Subject matter experts including bioethicists, principal investigators of genomic research projects and research administrators will be contacted. After de-duplication, titles and abstracts will be screened for eligibility. Data extraction will be undertaken using a piloted form designed in EPPI-Reviewer software before conducting risk of bias assessments by a pair of reviewers, acting independently. Any discrepancies will be resolved by consensus. Analysis will be done using a structured narrative synthesis and where feasible metanalysis. This review will attempt to highlight the

relevant data from this study will be made available upon study completion.

**Funding:** The systematic review is supported by the Fogarty International Center of the National Institutes of Health under Award Number D43TW010892 NS received the award No, the funders will not have a role in the study design The funding number is D43TW010892.

**Competing interests:** The authors have declared that no competing interest exists.

context of data sharing practices in the global North-South and South-South collaborative human genomic research in low- and middle-income countries. This review will enhance the body of evidence on ethical, legal and social implications of data sharing in international collaborative genomic research setting criteria for data sharing. The full report will be shared with relevant stakeholders including universities, civil society, funders, and departments of genomic research to ensure an adequate reach in low-and middle-income countries (LMICs).

## Background

### The problem of data sharing in genomics research

Data sharing is the practice of making research data available to other investigators conducting similar work. Data sharing is the practice of creating large databases by combining datasets into larger and separate datasets. The practice has become increasingly common by adding or combining research participants' data into larger repositories [1]. An increasing number of government departments, research communities, funding agencies and scholarly journals are developing initiatives and policies to promote data sharing and greater access to data, recognizing their enormous potential for scientific, social, and economic growth [2–5]. Genomic research is increasingly becoming common with individuals participating in large research projects being asked to share their data. Data sharing and research collaboration have become increasingly pervasive in the genomic research community [6]. This promises to increase research efficiency, expedite translational efforts of research results, and ensure the traceability and transparency of published studies and maximize utility of results [7–10].

The move to global data sharing has been facilitated by funding bodies, which have supported large international collaborative projects and developed open access policies to encourage wide-scale data sharing [11]. The data sharing policies of funders have shaped up and encouraged existing trends in scientific practice. Funders now require that data sharing be considered in every newly funded research project, unless there are justifiable reasons why this should not be so [12].

Fundamental to genomic research is the availability of data for research [13]. The need for broad access to genomic data brings along a host of ethical concerns, including those related to privacy and confidentiality, as well as fairness and equity in the conduct of human genomic collaborative research between researchers from high- income countries (HIC) and low-and middle income countries (LMICs) [13]. Open data policies from European countries [14, 15] and the United states of America [16] increasingly require custodians of others' genomic data to make it as widely available as feasible, including to researchers in other countries [17]. Data sharing is regarded as essential for enabling and promoting genomic research in a way that will maximize the benefits to public health [18].

Challenges of effective data sharing include 1) absence of established standards for data users, 2) researchers from low and middle income countries (LMICs) not appropriately credited and occasionally only recognized in the acknowledgment section of scientific publications, 3) the loss of intellectual property rights, and 4) the misuse of data [19, 20]. Analysis of national guidelines, policies and procedures from LMICs revealed major weaknesses and deficiencies in governance of genomic research and biobanking [21, 22]. There is lack of comprehensive ethico-legal frameworks to guide data sharing between HIC and LMIC. Discussions with

researchers over whether data providers should review results before publication, collaborate on the analysis, approve the analysis plan in advance, or limit conditions of data reuse continue to be important areas for dialogue on data sharing [20].

## Rationale for conducting this review on data sharing in genomic research

While data sharing is efficient and effective with considerable scientific and potential public health benefits, there are several pertinent ethical, legal and societal issues (ELSI) associated with data sharing in genomic research [13, 23–28]. One of the key challenges is determining how to protect the privacy of participants while enabling the sharing of data through global research networks [29]. Concerns about privacy and discrimination have led to policy responses from the National Human Genome Research Institute and additional policies from domestic and international countries that reaffirmed the recommendations for publicly sharing data [9]. These policies restrict access to genomic data as a means of protecting research participants, limit access to all genomic data, occasionally fail to respect the autonomy of these participants and, at the same time, unnecessarily limit the utility of the data [9]. Addressing these concerns while promoting genomic research, especially in Africa, requires stringent policies to guide the development of the governance frameworks for collaborative research. An understanding of the direction data sharing takes between continents, between countries within the global North-South and South-South and how they differ is crucial.

The proposed review is justifiable in comparison with a review of guidelines that was done in the conduct of genomic research in Africa [22]. While the objective of the review was to identify and characterize existing ethics-related guidelines and laws applicable to genetic and genomic research across much of Africa, the objective of our review is to identify perceptions and practices among a specific population (researchers, research participants and regulators) involved in the conduct of human genomic collaborative research in LMICs. The proposed review suggests a more robust methodological approach that will involve a systematic review of literature employing a narrative review approach in contrast with the review that involved only a documentary analysis. While there is an overlap with studies which looked at the ethical issues involved in data sharing in genomics research [29, 30], some of the recommendations proposed from these studies were how to respect and protect research participants when sharing data, and it is an area we hope to address with this review. There is also an overlap with studies [31, 32], in the systematic review methods which have been referenced.

With these policies, the question for many researchers is not whether to share data but how to. This creates several challenges for several areas of scientific practice. Yet, there is currently no synthesis of research evidence around data sharing in genomics research to guide decisions and practice in LMICs.

## Review objectives

The proposed work will investigate the context of data sharing practices in the global North-South and South-South collaborative human genomic research. Findings from this review could help in shaping the regulation of data sharing in collaborative human genomic research in LMICs.

Specifically, this review will:

1. Document the awareness, knowledge and perceptions of researchers, research participants, regulators and funders about data sharing in genomics research in LMICs.

2. Document the practices of researchers, research participants, regulators and funders about data sharing in genomics research in LMICs.

## Materials and methods

### Protocol development

The review will be protocol driven. It will be conducted according to the Preferred Reporting Items for Systematic Reviews and Meta-Analyses (PRISMA) checklist which refers to Open Science framework recommended for systematic reviews [33–36]. This protocol has been registered in PROSPERO (https://www.crd.york.ac.uk/prospero/) record CRD42022297984 and will be published in a peer-reviewed journal after further development.

### Review question

What are the data sharing perceptions and practices in human genomic collaborative research in LMICs?

Our review will be guided by the following elements of PICOST (population/setting, intervention/exposure, comparator, outcome, study design, timing of outcome assessment) (Table 1).

The review seeks to understand the context and extent of data sharing practices in the global North-South collaborative human genomic research in LMICs.

### Outcomes

The primary outcome of the review is data sharing. The secondary quantitative outcomes include: Proportion of projects that share data, Proportion of projects that share data with pre-agreements (Material Transfer Agreements, Data Sharing Agreements, Collaborative Agreements).

The secondary qualitative outcomes include: context, (types of data shared, how it is shared) data sharing practices, perceptions on data sharing, perceived barriers and facilitators to data sharing, ethical and legal implications, pros and cons of data sharing, implications of data sharing on uptake of findings.

**Table 1. Elements of population intervention context outcome setting time period (PICOST) for the review question.**

| Element | Description |
|---|---|
| Population | Researchers, research participants (children and adults), regulators and funders |
| Setting | Articles on data sharing in genomic research work in LMICs |
| Intervention/ exposure | Sharing raw data sets, analyzed data and intermediate data |
| Context | Global North–South and or South–South collaborative research projects |
| Outcomes | Primary outcome: Data sharing |
| | Secondary quantitative outcomes: Proportion of projects that share data, Proportion of projects that share data with pre-agreements (Material Transfer Agreements, Data Sharing Agreements, Collaborative Agreements) |
| | Secondary qualitative outcomes: context, (types of data shared, how it is shared) data sharing practices, perceptions on data sharing, perceived barriers and facilitators to data sharing, ethical social and legal implications, pros and cons of data sharing, collaboration |
| Study designs | Quantitative: Cross sectional surveys, case control, longitudinal study, cohort |
| | Qualitative: Descriptive studies, narrative synthesis, case studies, ethnography, grounded theory |
| | Mixed methods: Both qualitative and quantitative study designs |
| Time period | 2003–2022 (The year 2003 was chosen because lessons learned from the human genome project showed the importance of collaboration in genomic research) |

## Eligibility and selection of papers

**Screening of articles for inclusion.** All articles will be retrieved from data base searches and exported into endnote software for screening [33]. After de-duplication, all articles will be screened for by title and abstract. The full texts of all the papers that will be identified as potentially relevant will be retrieved by the lead reviewer (DES) with guidance of the professional Librarian.

**Inclusion criteria.** All studies that meet the PICOST criteria as per the research question will be included. Also, studies focused on data sharing, reported original research both published and grey literature will be included. Date restrictions will be applied to the initial electronic search to include articles published from 2003–2022, in English language.

The review will look at research conducted from 2003 when the human genome project ended. The year 2003 was chosen because lessons learned from the human genome project showed the importance of collaboration in genomic research [29, 37].

**Exclusion criteria.** Papers will be excluded if they focused on personal health records, clinical results, letters, articles that are not on data sharing or biobanking. Also, there will be language restrictions with papers not written in English excluded from the review. We will exclude papers that do not report relevant outcomes on data sharing and those that do not stratify results for LMICs.

**Data sources.** The electronic search shall be performed by the lead reviewer (DES) with the guidance of an Information Sciences Specialist (AK) who is a professional Librarian trained in systematic review methods, and an experienced librarian (DH) from Johns Hopkins University and through a review of existing databases and repositories including:

PubMed: https://www.ncbi.nlm.nih.gov/pubmed/

Google Scholar: https://scholar.google.com/

MEDLINE https: //www.nlm.nih.gov/bsd/medline.html

EMBASE: https://www.nlm.nih.gov/embase.html

Web of science: https://www.webofscience.html

**Electronic search strategy.** The electronic search terms will be generated and guided by the research question and the PICOST framework [38, 39]. The search strategy will be developed by the lead reviewer with guidance of senior Librarians. These key words and their synonyms will be combined using appropriate Boolean operators (AND, OR, NOT) in the electronic search engines across elements of PECOS (Population, Exposure, Comparison, Outcome and Study design). Truncation and wildcards will be added to terms where applicable. This search string will be piloted to validate in one data base and the validation will help establish availability of relevant studies which will help improve the sensitivity and specificity of the search string. This will be repeated for all the electronic data bases and refined in consultation with the co-authors. The following terms will be combined to search the PubMed database as follows:

(Researchers OR regulators OR participants) AND (Genomics OR genetic OR genetics OR human genome OR human genomics) AND (Data or data sharing or data link OR data sharing practices OR data breach OR data re-use) AND (low and middle income countries OR low and middle income country OR middle income country OR middle income population OR under developed country, OR lower income nations OR lower income populations OR underserved countries OR LMIC OR LMICs OR third world OR underserved population OR deprived country OR deprived population OR deprived populations OR poor country OR poor countries OR poor nation OR poor nations OR poor population OR poor populations OR poor world OR poorer countries OR poorer nations OR poorer population OR poorer populations OR developing economy OR developing economies OR less developed economy

OR less developed economies OR underdeveloped economies OR middle income economy OR middle income economies OR low income economy OR low income economies OR lower income economies OR low gdp OR low gnp OR low gross domestic OR low gross national OR lower gdp OR lower gross domestic OR lami country OR lami countries OR transitional country OR transitional countries OR emerging economies OR emerging nation OR emerging nations).

**Additional searches.** The bibliographies of included full text articles will be scanned for potentially eligible articles and subject matter experts from institutions in Uganda involved in genomic research at Makerere University, Uganda Virus Research Institute (UVRI) and Medical Research Council (MRC) will be contacted to identify institution reports, unpublished or ongoing studies on genomic research [32]. Websites of institutions that fund genomic work will be searched including that of the National Institute of Health (NIH), Medical Research Council (MRC) South Africa, Kenya Medical Research Institute (KEMRI) Welcome Trust, Research Institute and the United Kingdom Research and Innovation (UKRI). In addition, authors of included articles will be contacted for any relevant publications on the review topic [33].

**Minimizing bias in study identification and selection.** In order to minimize the risk of selection bias during abstraction in our systematic review conduct, a second reviewer will validate the electronic search in PubMed by performing an independent and duplicate search [32]. Similarly, the second reviewer will screen all articles excluded by the first reviewer. We will carry out independent study selection and data abstraction and the authorship team will resolve any differences by discussion and consensus.

**Data extraction and coding.** Data extraction will be done using a pre-designed and piloted form in Microsoft office excel [40] to capture the following information from included articles: author, year of study, country/region, study design and relevant outcomes [41]. The lead reviewer (DES) with one co-author will conduct data extraction in duplicate for a proportion of papers to enhance quality of the process. All disagreements will be resolved by consensus and/or discussion with the senior reviewer (EM) [35]. Results of the full text extraction will be shared with the remaining review authors to validate them.

**Assessment of quality.** The studies will be assessed for methodological quality using the Hawker checklist for reviewing disparate data systematically [42, 43]. Nine components will be assessed for methodological rigor which are: 1) Title and abstract, 2 introduction and aims, 3) method and data, 4) sampling, 5) data analysis, 6) ethics and bias, 7) results, 8) transferability or generalizability and 9) implications and usefulness [42]. This will be done with a possible score of 4 (good), 3 (fair), 2 (poor), or 1 (very poor). No studies will be eliminated based on quality criteria. Two authors will independently conduct this quality assessments any discrepancies will be resolved by consultation with the senior authors [43]. Decisions on acceptable levels of agreement will be based on the following cut-offs: poor < 0, slight, (0.0–0.2.), fair (0.21–0.40), moderate (0.41–0.60), substantial (0.61–0.80), and almost perfect (0.81–1.00) [43].

**Heterogeneity assessment.** This will only be done for quantitative studies assessing prevalence of knowledge about data sharing in genomic research. To assess the level of statistical heterogeneity in the studies, $I^2$ statistic will be used [44]. The $I^2$ statistic will indicate percentage (%) heterogeneity that can be attributed to between-study variance and interpreted: $I^2 =$ 25% (low), $I^2 =$ 50% (moderate), $I^2 =$ 75% (high). Sub group analysis will be conducted (low, moderate) [33].

**Data synthesis, coding and analysis.** Data synthesis will be done using a structured narrative approach. The analysis will employ the "best fit" framework synthesis that is commonly used for qualitative and mixed methods studies in by creating a *priori*-framework [45].

Quantitative studies will be summarized using descriptive statistics under selected subheadings, such as country, setting, focus, respondent characteristics, and the main themes identified [33, 43].

In the qualitative analysis, all documents that meet the inclusion criteria will be imported into NVivo 14 for analysis [22]. A code book will be created [46]. The codes will be organized and categorized based on published and non-published documents. The codes will have 5 broad themes including: 1) nature and extent of data sharing, 2) context of data sharing in genomic research, 3) barriers and facilitators for data sharing, 4) ethical and legal implications of data sharing and 5) pros and cons of data sharing. Descriptive themes or codes from primary studies will be summarized using the narrative approach and mapped on to a framework. The constant comparison approach will be used to identify differences and similarities across countries, types of studies and diagnostic groups where applicable [43]. Quotes from primary studies will be extracted and used as excerpts to illustrate themes as appropriate.

In triangulating the findings, an iterative and integrative approach of both qualitative and quantitative data will be employed during the interpretation phase [47, 48]. Differences in opinion will be resolved through discussions guided with the team.

**Handling of missing data.** Variables that are desired but missing or not reported will be denoted as not reported ('NR') and clarification will be sought by contacting the authors. No statistical methods will be employed for handling missing data [32].

**Risk of bias (ROB) of assessment of included studies.** We will assess for the risk of bias of included studies by adapting the Cochrane risk of bias tool for randomised and non-randomised studies [32, 49]. The ROB assessment will be done during the data extraction process. The risk of bias tool covers six domains of bias: selection bias, performance bias, detection bias, attrition bias, reporting bias, and other bias. Within each domain, assessments will be made for one or more items or outcomes [49]. For the observational studies, we will consider the following specific risk of bias aspects: similarity of baseline characteristics, sample size, control group, instrumental variables or potential confounding for all types of observational designs; questionnaire validity and reliability, sampling strategy, response rates for cross-sectional studies; attrition for cohort studies; choice of controls for case–control studies; analysis strategy, namely complete cases, per-protocol, as-treated or intention-to-treat; assessment for regression to the mean for controlled before and after or interrupted time-series designs [32]. ROB for qualitative studies will be assessed during analysis by employing a group-based approach of coding compared to an individual approach. Assessment of the methodological quality of the included systematic reviews will be done using the Joanna Briggs Institute (JBI) Critical Appraisal Checklist for Systematic Reviews and Research Syntheses' [31, 50].

Themes and quotes will be included during the abstraction process.

## Reporting and dissemination

We will align our reporting to the PRISMA statement [32, 51] which refers to open science framework [52] and share the full report with relevant stakeholders including universities, civil society, funders, and departments of genomic and genetic research to ensure an adequate reach especially in LMICs.

## Discussion

The current review will provide a context on data sharing practices in the global North-South; South-South partnerships and contribute to regulation of data sharing in collaborative human genomic research in LMICs. The findings of the study will contribute to the body of

knowledge on the ELSI of data sharing, and identifying ways of mitigating these ELSI concerns in international collaborative genomic research.

The findings of this review will guide regulators and policy makers in determining the best way on how to protect the personal interests of research participants while enabling the sharing of data through global research networks.

The findings from the study will also inform the debate around data sharing in genomic research and thus contribute to developing of appropriate models for data sharing in genomic collaborative research in these settings.

Systematic reviews, narrative reviews and overviews of reviews are relevant to guide practice and policy decisions as well as authors and readers of systematic reviews who ideally would use the findings in their work.

## Supporting information

**S1 Checklist. PRISMA-P 2015 checklist.**
(DOCX)

## Acknowledgments

We acknowledge the overall support and guidance provided by the principal investigator of the program Prof Nelson Sewankambo.

We acknowledge the support from Africa Center for Systematic Reviews with the protocol development, initial review and training on the use of EPPI-Reviewer.

We acknowledge the support and training from Donna Hesson a librarian at the Johns Hopkins University.

## Author Contributions

**Conceptualization:** Deborah Ekusai-Sebatta, Dickens Akena, Erisa Mwaka.

**Data curation:** Deborah Ekusai-Sebatta, Loyce Nakalembe, Robert Apunyo, Alison Annet Kinengyere, Eve Namisango, Ekwaro A. Obuku.

**Methodology:** Deborah Ekusai-Sebatta, Moses Ocan, David Kyaddondo, Ekwaro A. Obuku.

**Resources:** Alison Annet Kinengyere.

**Supervision:** Moses Ocan, Shenuka Singh, David Kyaddondo, Dickens Akena, Erisa Mwaka.

**Validation:** Eve Namisango.

**Visualization:** Deborah Ekusai-Sebatta, Robert Apunyo.

**Writing – original draft:** Deborah Ekusai-Sebatta.

**Writing – review & editing:** Deborah Ekusai-Sebatta, Moses Ocan, Shenuka Singh, David Kyaddondo, Dickens Akena, Alison Annet Kinengyere, Eve Namisango, Ekwaro A. Obuku, Erisa Mwaka.

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
