## [Decision Letter · Decision Letter 0]

17 May 2023

PONE-D-22-25350DATA SHARING PRACTICES IN COLLABORATIVE HUMAN GENOMIC RESEARCH IN LOW- AND MIDDLE-INCOME COUNTRIES: A SYSTEMATIC REVIEW PROTOCOLPLOS ONE

Dear Dr. Ekusai-Sebatta,

Thank you for submitting your manuscript to PLOS ONE. After careful consideration, we feel that it has merit but does not fully meet PLOS ONE’s publication criteria as it currently stands. Therefore, we invite you to submit a revised version of the manuscript that addresses the points raised during the review process.

We look forward to receiving your revised manuscript.

Kind regards,

Victoria E. Gibbon

Academic Editor

PLOS ONE

https://health-policy-systems.biomedcentral.com/articles/10.1186/s12961-017-0169-9

https://onlinelibrary.wiley.com/doi/10.1002/cl2.1161

https://www.annualreviews.org/doi/full/10.1146/annurev-genom-082410-101454

https://link.springer.com/article/10.1186/s12910-018-0310-5

In your revision ensure you cite all your sources (including your own works), and quote or rephrase any duplicated text outside the methods section. Further consideration is dependent on these concerns being addressed.

Reviewers' comments:

Reviewer's Responses to Questions

**Comments to the Author**

1. Does the manuscript provide a valid rationale for the proposed study, with clearly identified and justified research questions?

Reviewer #1: Yes

Reviewer #2: Yes

2. Is the protocol technically sound and planned in a manner that will lead to a meaningful outcome and allow testing the stated hypotheses?

Reviewer #1: Yes

Reviewer #2: Partly

3. Is the methodology feasible and described in sufficient detail to allow the work to be replicable?

Reviewer #1: Yes

Reviewer #2: Yes

4. Have the authors described where all data underlying the findings will be made available when the study is complete?

Reviewer #1: No

Reviewer #2: Yes

5. Is the manuscript presented in an intelligible fashion and written in standard English?

Reviewer #1: Yes

Reviewer #2: Yes

6. Review Comments to the Author

You may also provide optional suggestions and comments to authors that they might find helpful in planning their study.

Reviewer #1: "DATA SHARING PRACTICES IN COLLABORATIVE HUMAN GENOMIC RESEARCH IN LOW- AND MIDDLE-INCOME COUNTRIES: A SYSTEMATIC REVIEW PROTOCOL"

Since the completion of the Human Genome Project, numerous publications have highlighted the potential benefits of genomic data sharing and promoting accessibility of genomic data globally to facilitate genomic research. In fact, some funders are becoming prescriptive about the need for data sharing. There are, however, many challenges in data sharing, including the risk of privacy to study participants. Sub-Saharan Africa appears to be lagging in setting standards for data sharing in collaborative human genomic studies and there is a need to highlight the importance of regulating data sharing in LMICs and international collaborations which is what these authors aim to address. The PRISMA statements provide guidelines for reporting systematic reviews to ensure transparency, completeness, and accuracy of these reviews.

Here, Ekusai-Sebatta et al, designed a systematic review protocol to generate information which will highlight the context of data sharing in LMIC to policy makers and funders in Sub-Saharan Africa. The authors aim to investigate data sharing perceptions and practices amongst researchers, participants, and regulators in LMIC who are involved in human genomic collaborative research. These authors aim to conduct an electronic search of existing databases and repositories for published and grey literature focused on data sharing, using the recommended PRISMA-P checklist.

This is original work and has not been started yet. The authors have contextualised their proposed protocol along PRISMA-P guidelines. They have referenced relevant published work on the subject.

Major issues

The authors indicate that they have registered this protocol in PROSPERO and will share the full report with relevant stakeholders to ensure a wide reach. However, the authors do not indicate HOW they will disseminate the full report to accomplish this – I feel that this is important to specify in the protocol. PRISMA refers to the Open Science Framework (https://osf.io/r8a24/)

The authors also state that Data sharing is not applicable at this point of the study, however, it is a keyword for this protocol, and it is recommended that the authors indicate where/when the data generated by the electronic searches will be made accessible.

According to PLos guidelines: In the Data Availability Statement, briefly describe how data generated will be made accessible when the study is completed.

Risk of Bias:

ln 283 – is it the abstraction or extraction process?

ln 306- Ethics approval and Consent to participate – This subheading seems inappropriate for the text following it.

Reporting and Dissemination

Ln 285-287 – This paragraph is repeated almost verbatim in the Discussion Ln 302-305 and should preferably be reworded.

Other Issues

In the abstract, EMBASE and MEDLINE are not mentioned as data sources. Is there a reason for this?

Table 1: Should the word funders be included in Description of Population

Authors should consider adding intermediate data to the Description of intervention/ exposure.

Minimizing bias.

Ln 222- please clarify (i) whether this is extraction or abstraction.(ii) please indicate whether it is the authorship team who will resolve differences by discussion and consensus.

Ln 224- The subheading “Data abstraction and coding’’ should rather read ‘Data extraction and coding’’ because the paragraph discusses extraction.

Ln 229- Indicate who the senior reviewer is (EM)

Assessment of Quality

The authors should consider listing the ten components that will be assessed for methodological rigour.

Ln 234- This should read “No studies WILL BE eliminated…..” as this work is still to be done.

Data synthesis

Ln 254 – please reference ‘’a code book’’

Ln 264 – please indicate who will guide discussions to resolve the disagreement? My feeling is to avoid a negative word such as ‘disagreement’ – instead, you could say that ‘’differences of opinion will be resolved in guided team discussions’’. Indicate who will guide these sessions.

Other minor issues

Ln 52 – please clarify what is meant by ‘’óther’’ investigators

Ln 69- please indicate that the protocol is for data sharing of human genomic data

Ln 82- SSA never appears in full before this line.

Ln 111- ‘’with these policies……’’ does not follow from the previous sentence

Ln 121 and 123- be consistent throughout with the term genomic or genomics research

Reviewer #2: Dear Authors

I believe the systematic review of data sharing practices are indeed warranted in the era of genomics research. I am concerned however how much data the authors will be able to identify given that often details related to MTA or data sharing are not included in papers. HOWEVER, often this is implicit in the ethics obtained from specific institutions. Given that this information is not always included in papers how would the authors reconcile the data noted in publications to their objectives? I therefore suggest it is also important to understand the governance structures and legislation of the institutions and countries included in the papers to gain an understanding of potential practices of researchers in genomics. The assumption the authors are making includes that data is only going in one direction and it would be important to note the direction of data sharing between continents and within continents and how this differs.

My main recommendation is that authors include strategies to overcome some of the important missing data related to practice of data sharing and sharing of biological samples.

I look forward to seeing the data from this systematic review.

7. PLOS authors have the option to publish the peer review history of their article (what does this mean?). If published, this will include your full peer review and any attached files.

Reviewer #1: No

Reviewer #2: No

---

## [Author Response · Author response to Decision Letter 0]

7 Jul 2023

Data sharing practices in collaborative human genomic research in low- and middle-income countries: A systematic review protocol 

Editor: Please ensure that your manuscript meets PLOS ONE's style requirements,

Author: Thank you for observing this. The manuscript has been revised to ensure it meets PLOS ONE’S style requirements

Editor: We noticed you have some minor occurrence of overlapping text with the following previous publication(s), which needs to be addressed: In your revision ensure you cite all your sources (including your own works), and quote or rephrase any duplicated text outside the methods section. Further consideration is dependent on these concerns being addressed.

Author: While there is overlap, the study will address some of the recommendations made and all the studies have Yes, been referenced Lines (110-114)

Editor: We note that the grant information you provided in the ‘Funding Information’ and ‘Financial Disclosure’ sections do not match. When you resubmit, please ensure that you provide the correct grant numbers for the awards you received for your study in the ‘Funding Information’ section

Author: Thank you for catching this. I will ensure the correct grant number is provided both for the funding and financial disclosure

Reviwer 1: Ekusai-Sebatta et al, designed a systematic review protocol to generate information which will highlight the context of data sharing in LMIC to policy makers and funders in Sub-Saharan Africa. The authors aim to investigate data sharing perceptions and practices amongst researchers, participants, and regulators in LMIC who are involved in human genomic collaborative research. These authors aim to conduct an electronic search of existing databases and repositories for published and grey literature focused on data sharing, using the recommended PRISMA-P checklist.This is original work and has not been started yet. The authors have contextualised their proposed protocol along PRISMA-P guidelines. They have referenced relevant published work on the subject

Author: Thank you so much. The authors appreciate this positive feedback

Major issues

Reviewer 1: The authors indicate that they have registered this protocol in PROSPERO and will share the full report with relevant stakeholders to ensure a wide reach. However, the authors do not indicate HOW they will disseminate the full report to accomplish this – I feel that this is important to specify in the protocol. PRISMA refers to the Open Science Framework

Author: Thank you. We have included both in the abstract and the methods section that PRISMA refers to the Open Science Framework (lines 31, 135-136).

Author: The authors also state that Data sharing is not applicable at this point of the study, however, it is a keyword for this protocol, and it is recommended that the authors indicate where/when the data generated by the electronic searches will be made accessible.

According to PLOS guidelines: In the Data Availability Statement, briefly describe how data generated will be made accessible when the study is completed.

Author: The authors apologize for making such a grave error. In the data availability report, a statement has been included indicating that the systematic review was registered in PROSPERO and the registry number was included in their abstract (CRD42022297984) line 

32. 

Also, the authors have indicated under data sharing that “The search strategy is attached as an additional file. Additional data sources will be available on request. Requests can be made to the corresponding author at: ekusai@gmail.com” (Lines 341-343).

Reviewer 1: Risk of Bias: ln 283 – is it the abstraction or extraction process?

Author: Thank you. This has been changed to the data extraction process lines 229

Reviewer 1: ln 306- Ethics approval and Consent to participate – This subheading seems inappropriate for the text following it.

Author: Information has been included under the subheading Consent to participate and it now reads “There is no individual respondent data in the manuscript”. Lines 321

Reviewer1: Reporting and Dissemination

Ln 285-287 – This paragraph is repeated almost verbatim in the Discussion Ln 302-305 and should preferably be reworded. 

Author:Thank you for catching this. The repeated sentence has been deleted from the discussion

Other issues

In the abstract, EMBASE and MEDLINE are not mentioned as data sources. Is there a reason for this?

Author: Thank you. EMBASE and MEDLINE have been included in the abstract. (Lines 33-34)

Reviewer 1: Table 1: Should the word funders be included in Description of Population

Author: The word funders has been added to population (Table 1)

Reviewer 1: Authors should consider adding intermediate data to the Description of intervention/ exposure

Author: Thank you. Intermediate data has been added to the exposure (Table 1)

Reviewer 1: Minimizing bias.

Ln 222- please clarify (i) whether this is extraction or abstraction.(ii) please indicate whether it is the authorship team who will resolve differences by discussion and consensus

Author: Thank you. It is during abstraction. Yes, the authorship team will provide discussion and consensus. (Line 227).

Reviewer 1: Ln 224- The subheading “Data abstraction and coding’’ should rather read ‘Data extraction and coding’’ because the paragraph discusses extraction.

Author: Thank you.It is data is data extraction.

Reviewer 1: Ln 229- Indicate who the senior reviewer is (EM)

Author: The senior reviewer is EM (Line 234)

Reviewer: The authors should consider listing the ten components that will be assessed for methodological rigour.

Ln 234- This should read “No studies WILL BE eliminated…..” as this work is still to be done

Author: The nine components of the Hawker checklist have been listed. (lines 238-243). The nine components include: 1) Title and abstract, 2) introduction and aims, 3) method and data, 4) sampling, 5) data analysis, 6) ethics and bias, 7) results, 8) transferability, 9) implications and usefulness 

Reviewer 1: Data synthesis Ln 254 – please reference ‘’a code book’’

Author: A code book has been referenced (Line 261).

Reviewer 1: Ln 264 – please indicate who will guide discussions to resolve the disagreement? My feeling is to avoid a negative word such as ‘disagreement’ – instead, you could say that ‘’differences of opinion will be resolved in guided team discussions’’. Indicate who will guide these sessions.

Author: Thank you. This has been changed and now reads “Differences in opinion will be resolved through discussions guided with the team. (Lines 271-272).

Other minor issues 

Reviewer 1: 

Ln 52 – please clarify what is meant by ‘’óther’’ investigators

Author: Thank you. I have edited it and it now reads “Data sharing is the practice of making research data available to other investigators conducting similar work”.(Lines 53-54).

Reviewer 1: Ln 69- please indicate that the protocol is for data sharing of human genomic data

Author: Thank you. This has been indicated (Line 77)

Reviewer 1: Ln 82- SSA never appears in full before this line.

Author: This was an error. The word SSA has been removed (Line 83)

Reviewer 1: Ln 111- ‘’with these policies……’’ does not follow from the previous sentence

Author: This has been moved up to Lines immediately following previous sentence on policies (Lines 104-107)

Reviewer 1: Ln 121 and 123- be consistent throughout with the term genomic or genomics research

Author: Thank you. This has been corrected

Reviewer 2: 

I believe the systematic review of data sharing practices are indeed warranted in the era of genomics research. I am concerned however how much data the authors will be able to identify given that often details related to MTA or data sharing are not included in papers. HOWEVER, often this is implicit in the ethics obtained from specific institutions. Given that this information is not always included in papers how would the authors reconcile the data noted in publications to their objectives? I therefore suggest it is also important to understand the governance structures and legislation of the institutions and countries included in the papers to gain an understanding of potential practices of researchers in genomics. The assumption the authors are making includes that data is only going in one direction and it would be important to note the direction of data sharing between continents and within continents and how this differs. My main recommendation is that authors include strategies to overcome some of the important missing data related to practice of data sharing and sharing of biological samples.

Author: Thank you so much. You raise very important issues. While some of the information about data sharing maybe indeed be found in the methods section, not all researchers may write about it. The authors will look at the governance structures of the different institutions. 

Yes, indeed it is an important observation because since data sharing takes places between continents, knowledge of the direction is important. Also, through the review, we would like to find out how data sharing is happening between countries in the global south. A statement on this recommendation has been made in the protocol and now reads “An understanding of the direction data sharing takes between continents, within countries within the global south and how this differs is crucial” (Lines 101-102). Some of the strategies to overcome data sharing include contacting authors to reconcile some of the missing data and checking institutional websites.

Editor: Please ensure that your manuscript meets PLOS ONE's style requirements,

Author: Thank you for observing this. The manuscript has been revised to ensure it meets PLOS ONE’S style requirements

Editor: We noticed you have some minor occurrence of overlapping text with the following previous publication(s), which needs to be addressed: In your revision ensure you cite all your sources (including your own works), and quote or rephrase any duplicated text outside the methods section. Further consideration is dependent on these concerns being addressed.

Author: While there is overlap, the study will address some of the recommendations made and all the studies have Yes, been referenced Lines (110-114)

Editor: We note that the grant information you provided in the ‘Funding Information’ and ‘Financial Disclosure’ sections do not match. When you resubmit, please ensure that you provide the correct grant numbers for the awards you received for your study in the ‘Funding Information’ section

Author: Thank you for catching this. I will ensure the correct grant number is provided both for the funding and financial disclosure

Reviwer 1: Ekusai-Sebatta et al, designed a systematic review protocol to generate information which will highlight the context of data sharing in LMIC to policy makers and funders in Sub-Saharan Africa. The authors aim to investigate data sharing perceptions and practices amongst researchers, participants, and regulators in LMIC who are involved in human genomic collaborative research. These authors aim to conduct an electronic search of existing databases and repositories for published and grey literature focused on data sharing, using the recommended PRISMA-P checklist.This is original work and has not been started yet. The authors have contextualised their proposed protocol along PRISMA-P guidelines. They have referenced relevant published work on the subject

Author: Thank you so much. The authors appreciate this positive feedback

Major issues

Reviewer 1: The authors indicate that they have registered this protocol in PROSPERO and will share the full report with relevant stakeholders to ensure a wide reach. However, the authors do not indicate HOW they will disseminate the full report to accomplish this – I feel that this is important to specify in the protocol. PRISMA refers to the Open Science Framework

Author: Thank you. We have included both in the abstract and the methods section that PRISMA refers to the Open Science Framework (lines 31, 135-136).

Author: The authors also state that Data sharing is not applicable at this point of the study, however, it is a keyword for this protocol, and it is recommended that the authors indicate where/when the data generated by the electronic searches will be made accessible.

According to PLOS guidelines: In the Data Availability Statement, briefly describe how data generated will be made accessible when the study is completed.

Author: The authors apologize for making such a grave error. In the data availability report, a statement has been included indicating that the systematic review was registered in PROSPERO and the registry number was included in their abstract (CRD42022297984) line 

32. 

Also, the authors have indicated under data sharing that “The search strategy is attached as an additional file. Additional data sources will be available on request. Requests can be made to the corresponding author at: ekusai@gmail.com” (Lines 341-343).

Reviewer 1: Risk of Bias: ln 283 – is it the abstraction or extraction process?

Author: Thank you. This has been changed to the data extraction process lines 229

Reviewer 1: ln 306- Ethics approval and Consent to participate – This subheading seems inappropriate for the text following it.

Author: Information has been included under the subheading Consent to participate and it now reads “There is no individual respondent data in the manuscript”. Lines 321

Reviewer1: Reporting and Dissemination

Ln 285-287 – This paragraph is repeated almost verbatim in the Discussion Ln 302-305 and should preferably be reworded. 

Author:Thank you for catching this. The repeated sentence has been deleted from the discussion

Other issues

In the abstract, EMBASE and MEDLINE are not mentioned as data sources. Is there a reason for this?

Author: Thank you. EMBASE and MEDLINE have been included in the abstract. (Lines 33-34)

Reviewer 1: Table 1: Should the word funders be included in Description of Population

Author: The word funders has been added to population (Table 1)

Reviewer 1: Authors should consider adding intermediate data to the Description of intervention/ exposure

Author: Thank you. Intermediate data has been added to the exposure (Table 1)

Reviewer 1: Minimizing bias.

Ln 222- please clarify (i) whether this is extraction or abstraction.(ii) please indicate whether it is the authorship team who will resolve differences by discussion and consensus

Author: Thank you. It is during abstraction. Yes, the authorship team will provide discussion and consensus. (Line 227).

Reviewer 1: Ln 224- The subheading “Data abstraction and coding’’ should rather read ‘Data extraction and coding’’ because the paragraph discusses extraction.

Author: Thank you.It is data is data extraction.

Reviewer 1: Ln 229- Indicate who the senior reviewer is (EM)

Author: The senior reviewer is EM (Line 234)

Reviewer: The authors should consider listing the ten components that will be assessed for methodological rigour.

Ln 234- This should read “No studies WILL BE eliminated…..” as this work is still to be done

Author: The nine components of the Hawker checklist have been listed. (lines 238-243). The nine components include: 1) Title and abstract, 2) introduction and aims, 3) method and data, 4) sampling, 5) data analysis, 6) ethics and bias, 7) results, 8) transferability, 9) implications and usefulness 

Reviewer 1: Data synthesis Ln 254 – please reference ‘’a code book’’

Author: A code book has been referenced (Line 261).

Reviewer 1: Ln 264 – please indicate who will guide discussions to resolve the disagreement? My feeling is to avoid a negative word such as ‘disagreement’ – instead, you could say that ‘’differences of opinion will be resolved in guided team discussions’’. Indicate who will guide these sessions.

Author: Thank you. This has been changed and now reads “Differences in opinion will be resolved through discussions guided with the team. (Lines 271-272).

Other minor issues 

Reviewer 1: 

Ln 52 – please clarify what is meant by ‘’óther’’ investigators

Author: Thank you. I have edited it and it now reads “Data sharing is the practice of making research data available to other investigators conducting similar work”.(Lines 53-54).

Reviewer 1: Ln 69- please indicate that the protocol is for data sharing of human genomic data

Author: Thank you. This has been indicated (Line 77)

Reviewer 1: Ln 82- SSA never appears in full before this line.

Author: This was an error. The word SSA has been removed (Line 83)

Reviewer 1: Ln 111- ‘’with these policies……’’ does not follow from the previous sentence

Author: This has been moved up to Lines immediately following previous sentence on policies (Lines 104-107)

Reviewer 1: Ln 121 and 123- be consistent throughout with the term genomic or genomics research

Author: Thank you. This has been corrected

Reviewer 2: 

I believe the systematic review of data sharing practices are indeed warranted in the era of genomics research. I am concerned however how much data the authors will be able to identify given that often details related to MTA or data sharing are not included in papers. HOWEVER, often this is implicit in the ethics obtained from specific institutions. Given that this information is not always included in papers how would the authors reconcile the data noted in publications to their objectives? I therefore suggest it is also important to understand the governance structures and legislation of the institutions and countries included in the papers to gain an understanding of potential practices of researchers in genomics. The assumption the authors are making includes that data is only going in one direction and it would be important to note the direction of data sharing between continents and within continents and how this differs. My main recommendation is that authors include strategies to overcome some of the important missing data related to practice of data sharing and sharing of biological samples.

Author: Thank you so much. You raise very important issues. While some of the information about data sharing maybe indeed be found in the methods section, not all researchers may write about it. The authors will look at the governance structures of the different institutions. 

Yes, indeed it is an important observation because since data sharing takes places between continents, knowledge of the direction is important. Also, through the review, we would like to find out how data sharing is happening between countries in the global south. A statement on this recommendation has been made in the protocol and now reads “An understanding of the direction data sharing takes between continents, within countries within the global south and how this differs is crucial” (Lines 101-102). Some of the strategies to overcome data sharing include contacting authors to reconcile some of the missing data and checking institutional websites.

Editor: Please ensure that your manuscript meets PLOS ONE's style requirements,

Author: Thank you for observing this. The manuscript has been revised to ensure it meets PLOS ONE’S style requirements

Editor: We noticed you have some minor occurrence of overlapping text with the following previous publication(s), which needs to be addressed: In your revision ensure you cite all your sources (including your own works), and quote or rephrase any duplicated text outside the methods section. Further consideration is dependent on these concerns being addressed.

Author: While there is overlap, the study will address some of the recommendations made and all the studies have Yes, been referenced Lines (110-114)

Editor: We note that the grant information you provided in the ‘Funding Information’ and ‘Financial Disclosure’ sections do not match. When you resubmit, please ensure that you provide the correct grant numbers for the awards you received for your study in the ‘Funding Information’ section

Author: Thank you for catching this. I will ensure the correct grant number is provided both for the funding and financial disclosure

Reviwer 1: Ekusai-Sebatta et al, designed a systematic review protocol to generate information which will highlight the context of data sharing in LMIC to policy makers and funders in Sub-Saharan Africa. The authors aim to investigate data sharing perceptions and practices amongst researchers, participants, and regulators in LMIC who are involved in human genomic collaborative research. These authors aim to conduct an electronic search of existing databases and repositories for published and grey literature focused on data sharing, using the recommended PRISMA-P checklist.This is original work and has not been started yet. The authors have contextualised their proposed protocol along PRISMA-P guidelines. They have referenced relevant published work on the subject

Author: Thank you so much. The authors appreciate this positive feedback

Major issues

Reviewer 1: The authors indicate that they have registered this protocol in PROSPERO and will share the full report with relevant stakeholders to ensure a wide reach. However, the authors do not indicate HOW they will disseminate the full report to accomplish this – I feel that this is important to specify in the protocol. PRISMA refers to the Open Science Framework

Author: Thank you. We have included both in the abstract and the methods section that PRISMA refers to the Open Science Framework (lines 31, 135-136).

Author: The authors also state that Data sharing is not applicable at this point of the study, however, it is a keyword for this protocol, and it is recommended that the authors indicate where/when the data generated by the electronic searches will be made accessible.

According to PLOS guidelines: In the Data Availability Statement, briefly describe how data generated will be made accessible when the study is completed.

Author: The authors apologize for making such a grave error. In the data availability report, a statement has been included indicating that the systematic review was registered in PROSPERO and the registry number was included in their abstract (CRD42022297984) line 

32. 

Also, the authors have indicated under data sharing that “The search strategy is attached as an additional file. Additional data sources will be available on request. Requests can be made to the corresponding author at: ekusai@gmail.com” (Lines 341-343).

Reviewer 1: Risk of Bias: ln 283 – is it the abstraction or extraction process?

Author: Thank you. This has been changed to the data extraction process lines 229

Reviewer 1: ln 306- Ethics approval and Consent to participate – This subheading seems inappropriate for the text following it.

Author: Information has been included under the subheading Consent to participate and it now reads “There is no individual respondent data in the manuscript”. Lines 321

Reviewer1: Reporting and Dissemination

Ln 285-287 – This paragraph is repeated almost verbatim in the Discussion Ln 302-305 and should preferably be reworded. 

Author:Thank you for catching this. The repeated sentence has been deleted from the discussion

Other issues

In the abstract, EMBASE and MEDLINE are not mentioned as data sources. Is there a reason for this?

Author: Thank you. EMBASE and MEDLINE have been included in the abstract. (Lines 33-34)

Reviewer 1: Table 1: Should the word funders be included in Description of Population

Author: The word funders has been added to population (Table 1)

Reviewer 1: Authors should consider adding intermediate data to the Description of intervention/ exposure

Author: Thank you. Intermediate data has been added to the exposure (Table 1)

Reviewer 1: Minimizing bias.

Ln 222- please clarify (i) whether this is extraction or abstraction.(ii) please indicate whether it is the authorship team who will resolve differences by discussion and consensus

Author: Thank you. It is during abstraction. Yes, the authorship team will provide discussion and consensus. (Line 227).

Reviewer 1: Ln 224- The subheading “Data abstraction and coding’’ should rather read ‘Data extraction and coding’’ because the paragraph discusses extraction.

Author: Thank you.It is data is data extraction.

Reviewer 1: Ln 229- Indicate who the senior reviewer is (EM)

Author: The senior reviewer is EM (Line 234)

Reviewer: The authors should consider listing the ten components that will be assessed for methodological rigour.

Ln 234- This should read “No studies WILL BE eliminated…..” as this work is still to be done

Author: The nine components of the Hawker checklist have been listed. (lines 238-243). The nine components include: 1) Title and abstract, 2) introduction and aims, 3) method and data, 4) sampling, 5) data analysis, 6) ethics and bias, 7) results, 8) transferability, 9) implications and usefulness 

Reviewer 1: Data synthesis Ln 254 – please reference ‘’a code book’’

Author: A code book has been referenced (Line 261).

Reviewer 1: Ln 264 – please indicate who will guide discussions to resolve the disagreement? My feeling is to avoid a negative word such as ‘disagreement’ – instead, you could say that ‘’differences of opinion will be resolved in guided team discussions’’. Indicate who will guide these sessions.

Author: Thank you. This has been changed and now reads “Differences in opinion will be resolved through discussions guided with the team. (Lines 271-272).

Other minor issues 

Reviewer 1: 

Ln 52 – please clarify what is meant by ‘’óther’’ investigators

Author: Thank you. I have edited it and it now reads “Data sharing is the practice of making research data available to other investigators conducting similar work”.(Lines 53-54).

Reviewer 1: Ln 69- please indicate that the protocol is for data sharing of human genomic data

Author: Thank you. This has been indicated (Line 77)

Reviewer 1: Ln 82- SSA never appears in full before this line.

Author: This was an error. The word SSA has been removed (Line 83)

Reviewer 1: Ln 111- ‘’with these policies……’’ does not follow from the previous sentence

Author: This has been moved up to Lines immediately following previous sentence on policies (Lines 104-107)

Reviewer 1: Ln 121 and 123- be consistent throughout with the term genomic or genomics research

Author: Thank you. This has been corrected

Reviewer 2: 

I believe the systematic review of data sharing practices are indeed warranted in the era of genomics research. I am concerned however how much data the authors will be able to identify given that often details related to MTA or data sharing are not included in papers. HOWEVER, often this is implicit in the ethics obtained from specific institutions. Given that this information is not always included in papers how would the authors reconcile the data noted in publications to their objectives? I therefore suggest it is also important to understand the governance structures and legislation of the institutions and countries included in the papers to gain an understanding of potential practices of researchers in genomics. The assumption the authors are making includes that data is only going in one direction and it would be important to note the direction of data sharing between continents and within continents and how this differs. My main recommendation is that authors include strategies to overcome some of the important missing data related to practice of data sharing and sharing of biological samples.

Author: Thank you so much. You raise very important issues. While some of the information about data sharing maybe indeed be found in the methods section, not all researchers may write about it. The authors will look at the governance structures of the different institutions. 

Yes, indeed it is an important observation because since data sharing takes places between continents, knowledge of the direction is important. Also, through the review, we would like to find out how data sharing is happening between countries in the global south. A statement on this recommendation has been made in the protocol and now reads “An understanding of the direction data sharing takes between continents, within countries within the global south and how this differs is crucial” (Lines 101-102). Some of the strategies to overcome data sharing include contacting authors to reconcile some of the missing data and checking institutional websites.

Editor: Please ensure that your manuscript meets PLOS ONE's style requirements,

Author: Thank you for observing this. The manuscript has been revised to ensure it meets PLOS ONE’S style requirements

Editor: We noticed you have some minor occurrence of overlapping text with the following previous publication(s), which needs to be addressed: In your revision ensure you cite all your sources (including your own works), and quote or rephrase any duplicated text outside the methods section. Further consideration is dependent on these concerns being addressed.

Author: While there is overlap, the study will address some of the recommendations made and all the studies have Yes, been referenced Lines (110-114)

Editor: We note that the grant information you provided in the ‘Funding Information’ and ‘Financial Disclosure’ sections do not match. When you resubmit, please ensure that you provide the correct grant numbers for the awards you received for your study in the ‘Funding Information’ section

Author: Thank you for catching this. I will ensure the correct grant number is provided both for the funding and financial disclosure

Reviwer 1: Ekusai-Sebatta et al, designed a systematic review protocol to generate information which will highlight the context of data sharing in LMIC to policy makers and funders in Sub-Saharan Africa. The authors aim to investigate data sharing perceptions and practices amongst researchers, participants, and regulators in LMIC who are involved in human genomic collaborative research. These authors aim to conduct an electronic search of existing databases and repositories for published and grey literature focused on data sharing, using the recommended PRISMA-P checklist.This is original work and has not been started yet. The authors have contextualised their proposed protocol along PRISMA-P guidelines. They have referenced relevant published work on the subject

Author: Thank you so much. The authors appreciate this positive feedback

Major issues

Reviewer 1: The authors indicate that they have registered this protocol in PROSPERO and will share the full report with relevant stakeholders to ensure a wide reach. However, the authors do not indicate HOW they will disseminate the full report to accomplish this – I feel that this is important to specify in the protocol. PRISMA refers to the Open Science Framework

Author: Thank you. We have included both in the abstract and the methods section that PRISMA refers to the Open Science Framework (lines 31, 135-136).

Author: The authors also state that Data sharing is not applicable at this point of the study, however, it is a keyword for this protocol, and it is recommended that the authors indicate where/when the data generated by the electronic searches will be made accessible.

According to PLOS guidelines: In the Data Availability Statement, briefly describe how data generated will be made accessible when the study is completed.

Author: The authors apologize for making such a grave error. In the data availability report, a statement has been included indicating that the systematic review was registered in PROSPERO and the registry number was included in their abstract (CRD42022297984) line 

32. 

Also, the authors have indicated under data sharing that “The search strategy is attached as an additional file. Additional data sources will be available on request. Requests can be made to the corresponding author at: ekusai@gmail.com” (Lines 341-343).

Reviewer 1: Risk of Bias: ln 283 – is it the abstraction or extraction process?

Author: Thank you. This has been changed to the data extraction process lines 229

Reviewer 1: ln 306- Ethics approval and Consent to participate – This subheading seems inappropriate for the text following it.

Author: Information has been included under the subheading Consent to participate and it now reads “There is no individual respondent data in the manuscript”. Lines 321

Reviewer1: Reporting and Dissemination

Ln 285-287 – This paragraph is repeated almost verbatim in the Discussion Ln 302-305 and should preferably be reworded. 

Author:Thank you for catching this. The repeated sentence has been deleted from the discussion

Other issues

In the abstract, EMBASE and MEDLINE are not mentioned as data sources. Is there a reason for this?

Author: Thank you. EMBASE and MEDLINE have been included in the abstract. (Lines 33-34)

Reviewer 1: Table 1: Should the word funders be included in Description of Population

Author: The word funders has been added to population (Table 1)

Reviewer 1: Authors should consider adding intermediate data to the Description of intervention/ exposure

Author: Thank you. Intermediate data has been added to the exposure (Table 1)

Reviewer 1: Minimizing bias.

Ln 222- please clarify (i) whether this is extraction or abstraction.(ii) please indicate whether it is the authorship team who will resolve differences by discussion and consensus

Author: Thank you. It is during abstraction. Yes, the authorship team will provide discussion and consensus. (Line 227).

Reviewer 1: Ln 224- The subheading “Data abstraction and coding’’ should rather read ‘Data extraction and coding’’ because the paragraph discusses extraction.

Author: Thank you.It is data is data extraction.

Reviewer 1: Ln 229- Indicate who the senior reviewer is (EM)

Author: The senior reviewer is EM (Line 234)

Reviewer: The authors should consider listing the ten components that will be assessed for methodological rigour.

Ln 234- This should read “No studies WILL BE eliminated…..” as this work is still to be done

Author: The nine components of the Hawker checklist have been listed. (lines 238-243). The nine components include: 1) Title and abstract, 2) introduction and aims, 3) method and data, 4) sampling, 5) data analysis, 6) ethics and bias, 7) results, 8) transferability, 9) implications and usefulness 

Reviewer 1: Data synthesis Ln 254 – please reference ‘’a code book’’

Author: A code book has been referenced (Line 261).

Reviewer 1: Ln 264 – please indicate who will guide discussions to resolve the disagreement? My feeling is to avoid a negative word such as ‘disagreement’ – instead, you could say that ‘’differences of opinion will be resolved in guided team discussions’’. Indicate who will guide these sessions.

Author: Thank you. This has been changed and now reads “Differences in opinion will be resolved through discussions guided with the team. (Lines 271-272).

Other minor issues 

Reviewer 1: 

Ln 52 – please clarify what is meant by ‘’óther’’ investigators

Author: Thank you. I have edited it and it now reads “Data sharing is the practice of making research data available to other investigators conducting similar work”.(Lines 53-54).

Reviewer 1: Ln 69- please indicate that the protocol is for data sharing of human genomic data

Author: Thank you. This has been indicated (Line 77)

Reviewer 1: Ln 82- SSA never appears in full before this line.

Author: This was an error. The word SSA has been removed (Line 83)

Reviewer 1: Ln 111- ‘’with these policies……’’ does not follow from the previous sentence

Author: This has been moved up to Lines immediately following previous sentence on policies (Lines 104-107)

Reviewer 1: Ln 121 and 123- be consistent throughout with the term genomic or genomics research

Author: Thank you. This has been corrected

Reviewer 2: 

I believe the systematic review of data sharing practices are indeed warranted in the era of genomics research. I am concerned however how much data the authors will be able to identify given that often details related to MTA or data sharing are not included in papers. HOWEVER, often this is implicit in the ethics obtained from specific institutions. Given that this information is not always included in papers how would the authors reconcile the data noted in publications to their objectives? I therefore suggest it is also important to understand the governance structures and legislation of the institutions and countries included in the papers to gain an understanding of potential practices of researchers in genomics. The assumption the authors are making includes that data is only going in one direction and it would be important to note the direction of data sharing between continents and within continents and how this differs. My main recommendation is that authors include strategies to overcome some of the important missing data related to practice of data sharing and sharing of biological samples.

Author: Thank you so much. You raise very important issues. While some of the information about data sharing maybe indeed be found in the methods section, not all researchers may write about it. The authors will look at the governance structures of the different institutions. 

Yes, indeed it is an important observation because since data sharing takes places between continents, knowledge of the direction is important. Also, through the review, we would like to find out how data sharing is happening between countries in the global south. A statement on this recommendation has been made in the protocol and now reads “An understanding of the direction data sharing takes between continents, within countries within the global south and how this differs is crucial” (Lines 101-102). Some of the strategies to overcome data sharing include contacting authors to reconcile some of the missing data and checking institutional websites.

I have attched a point by point response to the reviewers comments

---

## [Decision Letter · Decision Letter 1]

4 Oct 2023

DATA SHARING PRACTICES IN COLLABORATIVE HUMAN GENOMIC RESEARCH IN LOW- AND MIDDLE-INCOME COUNTRIES: A SYSTEMATIC REVIEW PROTOCOL

PONE-D-22-25350R1

Dear Dr. Ekusai-Sebatta,

We’re pleased to inform you that your manuscript has been judged scientifically suitable for publication and will be formally accepted for publication once it meets all outstanding technical requirements.

Kind regards,

Victoria E. Gibbon

Academic Editor

PLOS ONE

Additional Editor Comments (optional):

There are some minor suggestions in the proof from the second reviewer.

Reviewers' comments:

Reviewer's Responses to Questions

**Comments to the Author**

1. Does the manuscript provide a valid rationale for the proposed study, with clearly identified and justified research questions?

Reviewer #2: Yes

2. Is the protocol technically sound and planned in a manner that will lead to a meaningful outcome and allow testing the stated hypotheses?

Reviewer #2: Yes

3. Is the methodology feasible and described in sufficient detail to allow the work to be replicable?

Reviewer #2: Yes

4. Have the authors described where all data underlying the findings will be made available when the study is complete?

Reviewer #2: Yes

5. Is the manuscript presented in an intelligible fashion and written in standard English?

Reviewer #2: Yes

6. Review Comments to the Author

You may also provide optional suggestions and comments to authors that they might find helpful in planning their study.

Reviewer #2: I believe the authors have addressed all concerns and matters raised in the first review process. I have nothing further to add.

7. PLOS authors have the option to publish the peer review history of their article (what does this mean?). If published, this will include your full peer review and any attached files.

Reviewer #2: No

---

## [Editor Report · Acceptance letter]

23 Oct 2023

PONE-D-22-25350R1 

Data sharing practices in collaborative human genomic research in low- and middle-income countries: A systematic review protocol 

Dear Dr. Ekusai-Sebatta:

I'm pleased to inform you that your manuscript has been deemed suitable for publication in PLOS ONE. Congratulations! Your manuscript is now with our production department. 

Kind regards, 

on behalf of

Prof Victoria E. Gibbon 

Academic Editor

PLOS ONE